# A Study on Pavement Classification and Recognition Based on VGGNet-16 Transfer Learning

Junyi Zou , Wenbin Guo and Feng Wang *

School of Automotive and Traffic Engineering, Wuhan University of Science and Technology,
Wuhan 430065, China; zoujunyi@wust.edu.cn (J.Z.); guowenbin19971021@163.com (W.G.)
* Correspondence: wangfeng95407@163.com

**Abstract:** The types of road surfaces on which intelligent connected cars operate are complicated and varied, and current research lacks the achievement of real-time and reasonably high accuracy for road surface categorization. In this research, we provide a deep learning-based technique for classifying and identifying road surfaces that makes use of an improved (VGGNet-16) model, in conjunction with a transfer learning strategy, to gather data from the road surface in front of the car using an on-board camera. To accurately classify data based on obtained road surface photos, the dataset is first preprocessed, then pretrained weights are frozen, and the network is initialized using transfer learning parameters. In order to explore the accuracy analysis of the various models regarding the identification of six types of road surfaces, comparisons were made via the VGG16, AlexNet, InceptionV3, and ResNet50 models, using the same parameter values. The experimental findings demonstrate that the improved VGGNet-16 model, combined with the transfer learning approach, achieves 96.87% accuracy for the classification and recognition of pavements, demonstrating the improved network model's superior accuracy for these tasks. Additionally, the driving recorder of the vehicle may be used as the sensor to complete pavement detection, which has significant financial advantages.

**Keywords:** image classification; pavement classification recognition; VGGNet-16 transfer learning; deep learning

## 1. Introduction

Safety has been the top priority in the development of intelligent connected vehicles, due to the rising numbers of personal vehicles and traffic accidents, while autonomous driving technology offers a very good chance of lowering traffic accidents, easing traffic congestion, etc. It is a very well-supported field of study in terms of imaging and image processing; recognizing the state of the road may significantly lower the frequency of accidents. The extraction of road surface state data is impacted by elements such as light levels and bumps due to vehicle motion, which leads to huge errors in the dataset. Driving conditions are affected by the weather, and the adhesion coefficient of the car varies when driving on different road surfaces. Although the traditional image recognition classification methods can extract textural information regarding the road surface features in order to classify the road surface, the accuracy rate will be low due to the influence of data acquisition, and the adaptability will also be relatively low, so there are still many areas in road surface recognition that need improvement.

Traditional machine learning is employed in many current pavement condition recognition and detection techniques, such as the support vector machine (SVM) [1], KNN [2], the parsimonious Bayes technique [3], decision trees [4], and the artificial neural network (ANN), which are used to categorize the extracted pavement data in terms of dryness and humidity. Numerous researchers in the field of machine learning have studied pavement identification extensively and have produced numerous findings. Liu et al. [5] used SVM

to study pavements in wet conditions, and selected kernel functions and classifiers applicable to pavement recognition to enable recognition in wet conditions. However, the method's accuracy rate was low, while the matrix operation occupied a large amount of the training time and was relatively inefficient. Li [6] developed an icy road surface detection system by acquiring the texture and color of a high-speed road surface, extracting texture features from the acquired road surface, and applying an SVM classifier to identify the icy road sections. The accuracy of the identified road image achieved 80.4 percent, but the relatively large amount of effort required to acquire the image at the terminal can cause the preparation of surface recognition to become complicated, making it less efficient. Jian Wan [7] proposed an experimental system for determining slippery road surfaces using machine vision, with an accuracy of 70% to 80%. When selecting image color features and describing features, it was difficult to ensure their computation in real time. Li et al. [8] proposed obtaining the average construction depth of pavement texture parameters, based on tilt photography technology combined with positioning technology, to achieve real-time feedback on the skid resistance of the pavement, but the accuracy of its recognition was affected by the lack of light on the acquired pavement images, which needed to be enhanced at the time of image acquisition and for other work. Uriana, Bai et al. [9] proposed a comprehensive model for automatic pavement damage detection and recognition using a deep learning approach. The accuracy level of this model achieved 97% for localization and 92.4% for classification, showing that the accuracy of the classification part of their model algorithm needs to be improved. Yang et al. [10] designed a residual neural network-based algorithm for pavement wet conditions, achieving an accuracy of 85.4% for recognition. Compared to the classification functions of deep learning, the accuracy and classification speed of the selected classifier of SVM needs to be improved. With the rising popularity and development of deep learning in recent years, the processing of image data has been continuously improving, with higher speed and accuracy than traditional machine learning. Deep learning has also achieved much in terms of object recognition and target detection, as well as in the aerospace and medical fields [11–15]. The convolutional neural network is one of the most representative algorithms in the field of deep learning [16]; there are four classical networks in the field of convolutional neural network image classification, comprising AlexNet, VGGNet, Google Inception Net, and ResNet. The VGGNet network model used in this paper was proposed by Simonyan and Zisserman of Oxford University in 2014, who presented a deep convolutional neural network [17]. It is clear from the literature [18] that the VGGNet-16 model is more suitable for tuning parameters, and the model in question can be constructed using the appropriate improvements. Zhang et al. [19] employed a machine learning method to evaluate the roughness of the road surface. The category of road roughness was estimated from the inertial sensor on the vehicle, then the dynamic model of the vehicle suspension system and the mathematical simulation of the road roughness curve were combined to verify the success of the method, which provided some features for the machine learning. The accuracy of the roughness classification of the road surface in this paper was only 90%, showing that there is room for improvement in terms of real-time performance. Llopis-Castelló et al. [20] identified and classified a variety of road surface issues, adopting a lightweight convolutional neural network, and achieved their highest accuracy of 92.35% through 1000-iteration pre-training; they also classified the different conditions of urban road surfaces. The method described in their paper can greatly reduce the cost and time required for a visual assessment of road surface conditions. Doycheva et al. [21] trained a large amount of data on GPU to solve the problem that the CPU could not analyze pavement images in real time. In their paper, an automatic road condition detection method was proposed, wherein a real-time noise removal, background correction, and pavement condition detection method was employed by the graphics processing unit, and the description values of classification were calculated using wavelet transform. This method can be used for real-time image preprocessing and analysis. Georgios M. Hadjidemetriou and Symeon E. Christodoulou [22] designed a road management system to ensure effective on-road function and the safety

of passengers in vehicles. This paper introduces a vision-based system that uses low-cost technology to detect damaged areas of the road surface, then uses image entropy and image processing power to identify video frames, with an accuracy of 89.2%. The system can improve the efficiency of road repair for the transportation department and save significant costs. After hyperparameter tuning and validation of the VGGNet-16 model, this paper proposes to combine transfer learning [23] with the improved VGGNet-16 model to obtain a network model suitable for pavement condition identification. Real-time classification and recognition of dry asphalt pavement, wet asphalt pavement, and snow and ice on the pavement were performed on the pavement test dataset and compared with VGG16, AlexNet, InceptionV3, and ResNet50 models [24–27], using the images acquired by the vehicle camera as inputs and classifying the different pavement types. This can provide pre-scanning data for the subsequent control of dynamics information warnings.

The rest of the paper is organized as follows. In Section 2, the relevant literature and the applications of the VGGNet-16 model are introduced, the training process of transfer learning is explained, the improvement method is discussed, and the network model of the proposed VGGNet-16 transfer learning method is constructed. The pre-processing and empirical validation sections of our dataset are given in Section 3, as well as a comparison of parallax, weight analysis, and accuracy. A discussion of the results obtained in Section 3 is given in Section 4. In addition, we show the subsequent work design flow chart of this paper's recognition classification algorithm in Figure 1.

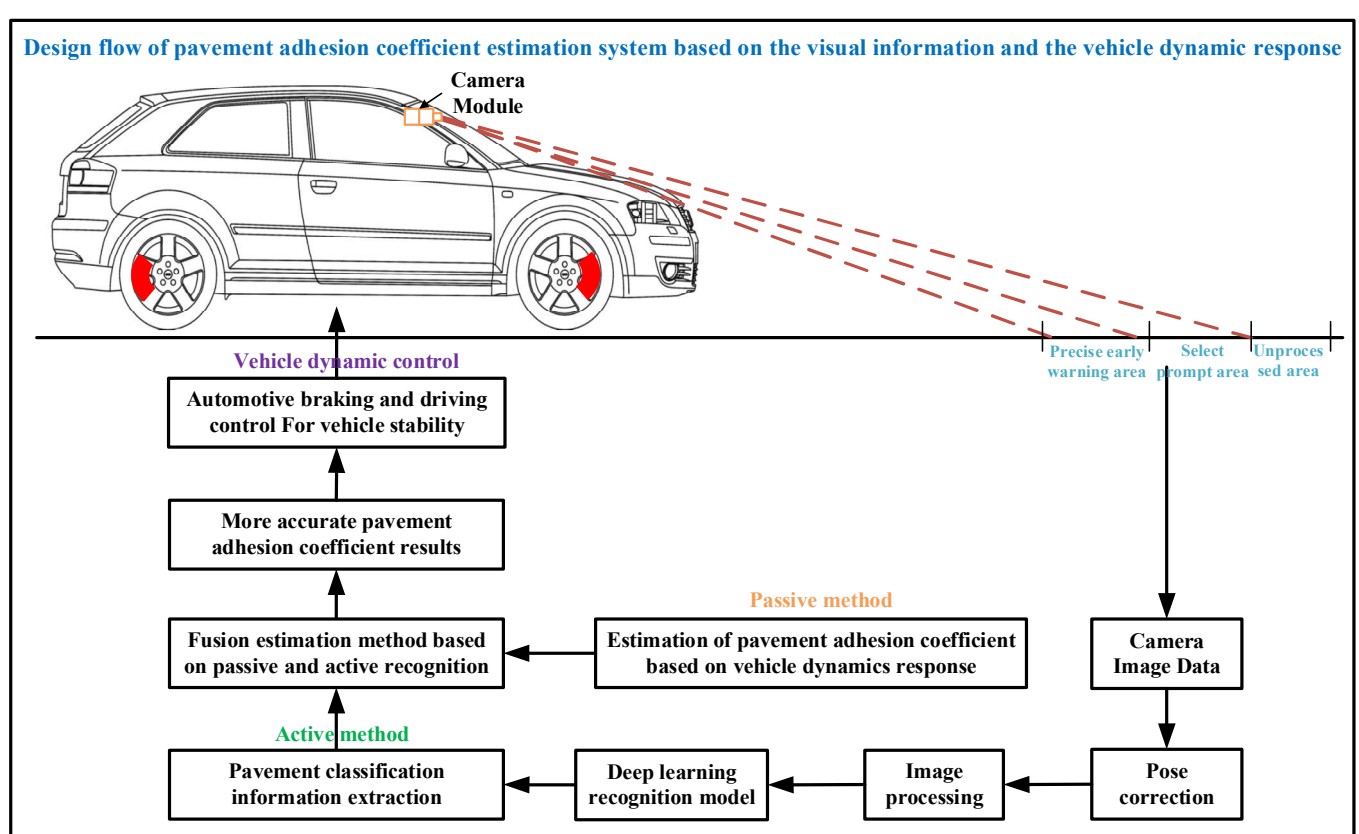

**Figure 1.** A design flowchart of the pavement adhesion coefficient estimation system.

The research in this paper addresses the primary task of designing a roadway adhesion coefficient estimation system, based on visual information and the vehicle's dynamic response. The design flowchart of the system is shown in Figure 1, above. First, we divide the roadway obtained from the onboard vision sensor into three zones, which comprise the precise warning cueing zone, the selective warning cueing zone, and the unprocessed cueing zone. According to our recognized road ahead prompts for safety warnings, we then

load the road surface information obtained by the on-board vision sensor into the domain controller after position correction and image pre-processing, using the deep learning model in this paper to assess the road ahead of the vehicle for the extraction of classification information. Then, based on the dynamic response of the vehicle's passive estimation of the road surface adhesion coefficient and passive and active recognition using the Fusion estimation method, the fused road surface attachment coefficient estimation will yield more accurate estimation results, so that the stability of the vehicle can be further controlled while also improving driving safety.

## 2. Materials and Methods

VGGNet-16 is a model proposed by the Visual Geometry Group at the University of Oxford [28], which achieved excellent results in the ImageNet Image Classification and Localization Challenge 2014 ILSVRC-2014, placing second in the classification task and first in the localization task. VGG represents a good successor to AlexNet, with effective discriminative features, and the network is much deeper.

The main principle of the VGGNet model is to use smaller convolution kernels, with the aim of reducing the number of parameters raised. This is achieved by an increase in convolutional layers and a reduction in the weight space. The main body of the model has the following structures: the VGGNet-16 and the VGGNet-19 [29,30].

VGGNet-16 is derived from the AlexNet model [31]; the main body of the VGGNet-16 network model [32–34] consists of five groups of convolutional layers and three groups of completely connected layers with a SoftMax activation function. Each group of convolutional layers is separated by a maximum pooling layer and the activation functions of all its hidden layers use the ReLU function. The VGGNet-16 uses a convolutional kernel size of (3 × 3) and a maximum pooling layer size of (2 × 2). The use of tiny convolutional kernels is preferable to using large convolutional kernels because such multilayer nonlinear layers can increase the depth of the network model. The use of narrow convolutional kernels in VGGNet-16 is intended to both increase the depth of the network model and improve the training results, while preserving the perceptual domain. The first two layers of the completely connected layer of VGGNet-16 have 4096 neurons, which generate a large number of feature parameters at the end of training. Therefore, the number of neurons is reduced in subsequent refinements to improve the network model, and this improvement helps to prevent overfitting of the model and can reduce the weight space of the model. The structure diagram of the VGGNet-16 network model is shown in Figure 2. The parameter information is given in Table 1.

**Table 1.** The parameter information.

| ConvNet Configuration | | | | | |
|---|---|---|---|---|---|
| A | A-LRN | B | C | D(VGGNet-16) | E(VGGNet-19) |
| 11 weight layers | 11 weight layers | 13 weight layers | 16 weight layers | 16 weight layers | 19 weight layers |
| Input (RGB image) | | | | | |
| Conv3-64 | Conv3-64 LRN | Conv3-64 | Conv3-64 | Conv3-64 | Conv3-64 |
| | | Conv3-64 | Conv3-64 | Conv3-64 | Conv3-64 |
| MaxPool | | | | | |
| Conv3-128 | Conv3-128 | Conv3-128 | Conv3-128 | Conv3-128 | Conv3-128 |
| | | Conv3-128 | Conv3-128 | Conv3-128 | Conv3-128 |
| Maxpool | | | | | |
| Conv3-256 | Conv3-256 | Conv3-256 | Conv3-256 | Conv3-256 | Conv3-256 |
| Conv3-256 | Conv3-256 | Conv3-256 | Conv3-256 | Conv3-256 | Conv3-256 |
| | | | Conv1-256 | Conv3-256 | Conv3-256 |
| | | | | | Conv3-256 |

**Table 1.** *Cont.*

| ConvNet Configuration | | | | | |
|---|---|---|---|---|---|
| | | | Maxpool | | |
| Conv3-512<br>Conv3-512 | Conv3-512<br>Conv3-512 | Conv3-512<br>Conv3-512 | Conv3-512<br>Conv3-512<br>Conv3-512 | Conv3-512<br>Conv3-512<br>Conv3-512 | Conv3-512<br>Conv3-512<br>Conv3-512<br>Conv3-512 |
| | | | Maxpool | | |
| Conv3-512<br>Conv3-512 | Conv3-512<br>Conv3-512 | Conv3-512<br>Conv3-512 | Conv3-512<br>Conv3-512<br>Conv1-512 | Conv3-512<br>Conv3-512<br>Conv3-512 | Conv3-512<br>Conv3-512<br>Conv3-512<br>Conv3-512 |
| | | | Maxpool<br>FC4096<br>FC4096<br>FC1000<br>SoftMax | | |

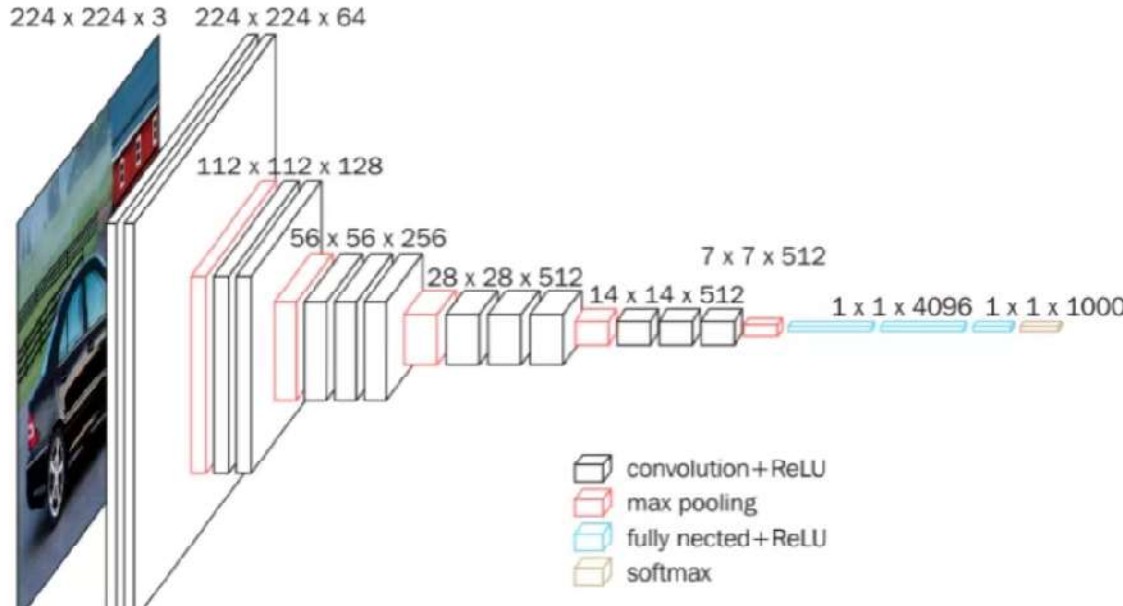

**Figure 2.** The VGGNet-16 network model.

The activation functions of the six network models described in the above table are all ReLU activation functions, of which the D structure is the "VGGNet-16" and the E structure is the "VGGNet-19" used in this paper. The difference between the two networks is that there are three additional convolutional layers; also, the Leaky ReLU activation function is used in this paper instead of the ReLU function in the original model, as will be explained in detail in the next sections.

### 2.1. VGGNet-16 Network Model Training Process

For pavement classification discrimination in the transfer learning used in this paper, the corresponding dataset, *T*, can be expressed as:

$$T = \left\{ (x^1, y^1), (x^2, y^2), \ldots, (x^N, y^N) \right\} \tag{1}$$

where *N* is the total number of training datasets, and *x*, *y* are the input and output of each image. Different pavement states will correspond to different labeled outputs. The training process of the improved algorithmic network model, based on VGGNet-16, is as follows:

(1) The VGGNet-16 network reads the virtual pavement training dataset $(x, y)$ and inputs the $x^i$ of each image in the virtual dataset into the input layer of the network model. After propagating the designed network forward, the output result obtained in the output layer of the model is called the predicted output, $\hat{y}$;

(2) The predicted output, $\hat{y}$, is then compared with the desired output, $y$. In classification problems, the cross-entropy loss function $J(y, \hat{y})$ is often used to represent the difference between the predicted and desired outputs.

$$y = (y_1, y_2, \ldots, y_N)^T \tag{2}$$

$$\hat{y} = (\hat{y}_1, \hat{y}_2, \ldots, \hat{y}_N)^T \tag{3}$$

$$J(y, \hat{y}) = -\sum_{i=1}^{N} [y_i In(\hat{y}_i)] = -y^T \cdot In(\hat{y}) \tag{4}$$

In the above equation, $N$ denotes the total number of neurons in the output layer, i.e., the total number of categories in the dataset labels.

(3) The $J$ obtained from the data image input to the neural network is often a large number, which means that there is a certain deviation between $J$ and 0. This deviation is passed from the output layer to the input layer through the backpropagation (BP) algorithm [35], in which the weights and biases in the network model are finely adjusted, based on the gradient, to reduce the value of $J$.

(4) Then, according to the parameter gradient obtained via the backpropagation algorithm, the convolutional neural network model will update the parameters in the model, based on this gradient descent method, as a way to reduce the value of the loss function. The process of updating the parameters of the model parameter $\theta(w,b)$ is as follows.

$$\theta_{k+1} = (\theta_k) - \alpha \frac{\partial J}{\partial \theta}(\theta_k) \tag{5}$$

In the above equation, $\theta_k$ is the value of the parameter obtained at the $k$th iteration in the model training, and $\alpha$ is the learning rate of the parameter $\theta$.

(5) At this point, one learning or training process of the neural network model is now complete. The next step is to repeat the forward and backward propagation process as many times as necessary until the end of training.

### 2.2. Transfer Learning Process

As more and more machine learning application scenarios have emerged, one after another, in recent years, and the existing supervised learning techniques that perform better require a large volume of labeled data, which is a huge task, transfer learning [36] has received more and more research attention. In this paper, we propose a combination of transfer learning and an improved VGGNet-16 network model for the classification and recognition of pavement types. The selected improved VGGNet-16 model was extensively trained on the training dataset, then the subsequent test and validation sets were subjected to transfer learning with the improved model. We show the flow of transfer learning in Figure 3.

### 2.3. Improvement of the Activation Function

The different introduction methods of activation functions in the network model have a profound impact on the training of the model. In the VGGNet-16 model, the activation function of all its hidden layers uses the ReLU function as the activation function, which is mathematically defined as:

$$f(x) = \max(0, x). \tag{6}$$

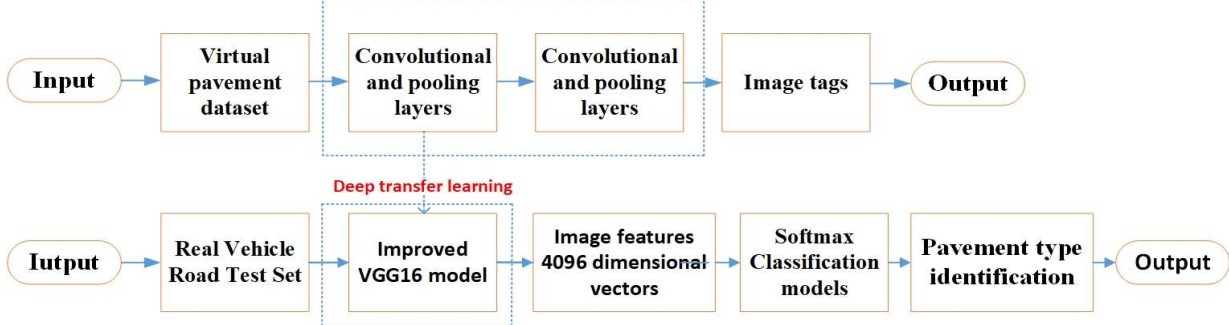

**Figure 3.** Transfer learning process for pavement classification recognition, based on the improved VGGNet-16.

First, the input pavement data will generate numerous negative values after the convolution operation, then the ReLU function will modify the negative values to 0. This loses some of the feature information from the dataset. The Leaky ReLU activation function outputs a different value from the ReLU function, depending on the input value. If the input feature information is positive, the Leaky ReLU function output is the same as the ReLU function. If the input feature information is negative, the Leaky ReLU function outputs a slightly negative value, defined by $\alpha z$ (where $\alpha$ is a tiny value; this value is defined as 0.02 in this paper, which is the feature input value). Its mathematical definition is as follows:

$$f(x) == \begin{cases} x, x > 0 \\ \alpha x, x \le 0 \end{cases}. \tag{7}$$

Under the same conditions, the Leaky ReLU activation function is used to replace the ReLU function, and the accuracy of the training and test sets is improved by 3.27% and 4.35%, respectively. In Section 3.4, we will show the effects of two activation functions on the performance of the network model.

*2.4. Introduction of Residual Structure*

In the original model of the VGGNet-16 network presented in this paper, the residual structure is missing. As the network deepens, the training errors first decrease and then increase; their errors are not due to overfitting but to the difficulty of training due to the deeper layers of the network. In order to solve the above problem, this paper constructs a residual block class containing two convolutional layers, a Leaky ReLU activation function, and a BN algorithm [36] before the activation function to normalize the input data. The residual structure is shown in Figure 4. The convolution and pooling layers in VGGNet-16 can extract the low-level features of the image, while the inclusion of a residual structure can learn higher-level features. The inclusion of the residual structure allows for the learning of higher-level feature representations and is intended to address the impact of gradient disappearance and network depth on learning performance. To some extent, the inclusion of the three residual structures can avoid the loss of features in the convolutional layer when performing image information transfer, and new image features can be learned on the basis of the input features to prevent gradient disappearance due to the deep network layers of VGGNet-16.

*2.5. Fill Dropout Inactive Layer*

Srivastava et al. proposed a dropout inactive layer in 2012. In the forward propagation process of a training dataset, some neurons are randomly selected so that the activation values of neurons must be suspended according to a certain probability, to enhance the generalization ability of the network model and avoid overfitting of the network.

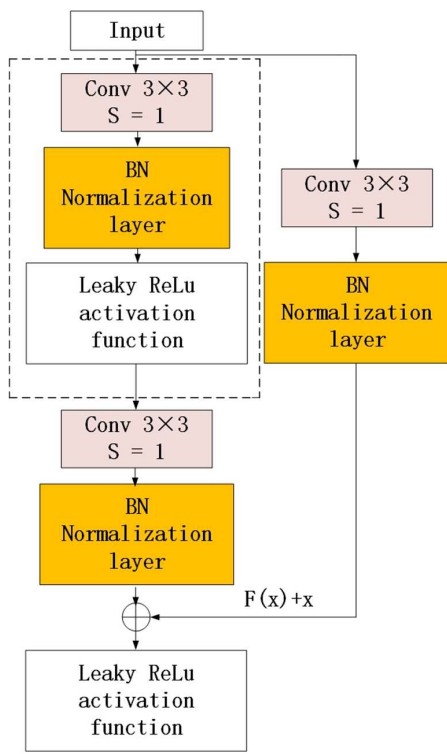

**Figure 4.** Residual structure.

*2.6. Model Building for VGGNet-16 Transfer Learning*

The proposed VGGNet-16 transfer learning-based pavement recognition system takes the approach of obtaining pavement information from the vehicle vision sensors, classifying and recognizing it as follows.

(1) Image acquisition: Based on the vehicle vision sensor used to detect the road surface traveled by the car in real time, the characteristic information of the road surface is accepted and released in the domain controller after pre-processing measures such as light enhancement.

(2) VGGNet-16 transfer model construction: In this paper, we introduce the Leaky ReLU activation function to replace the ReLU activation function in the original model and fill in the residual structure and dropout inactive layer, etc., to improve the network structure of VGGNet-16 and adjust the parameters. The pre-trained model is then saved from the input layer to the Block4_conv3 layer. The ConvBNLR module consists of a convolutional layer and a normalization layer with a Leaky ReLU activation function, where the convolutional kernel size is set to the feature matrix with a step size of 1. The overall model consists of two ConvBNLR modules, three residual modules, a Maxpooling layer for image size reduction, and a sufficiently connected layer to output three classifications using the Softmax function for classification. The improved VGGNet-16 model reduces the number of parameters in the network model by successive stacks of convolutional kernels with the size of the convolutional kernel, while ensuring the same perceptual field. Then, it increases the number of nonlinear units in the model by adding a residual structure after the VGG module using jump connections, which solves the gradient disappearance and improves the performance of the network's deep learning. The parameters of the original model of VGGNet-16 are concentrated in the fully connected layer, in order to prevent overfitting, improve the classification accuracy of the pavement, and accelerate the convergence of the network model. After freezing the pre-trained weights, a global average pooling layer, a flattened layer, two dropout inactive layers, and three fully connected layers are added in turn. The input data training set is standardized by uniform

size [0.5, 0.5, 0.5] processing, using the Leaky ReLU function after the connection layer and the last classifier using the Softmax function (for the output category of 3 classes of pavement), outputting 3 categories and labeling the categories as 0 for wet asphalt pavement, 1 for dry asphalt pavement, 2 for snow and ice pavement, 3 for concrete pavement output, 4 for gravel pavement output, and 5 for jointed pavement output. The modified VGGNet-16 network model and the technical flow of this research paper are shown in Figure 5.

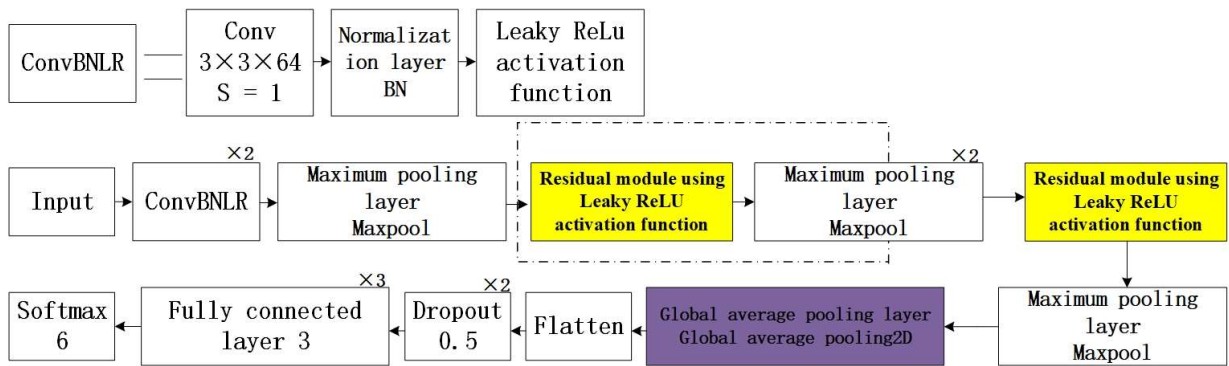

**Figure 5.** VGGNet-16 transfer learning model.

(3) Pavement recognition decision: This is calculated according to the pavement acquisition image, combined with the data pre-training weights, based on a deep learning transfer model, which is used to complete the construction of the pavement classification recognition model.

The road surface recognition system consists mainly of in-vehicle vision sensors and intelligent driving domain controllers, the detection process of which is illustrated in Figure 6.

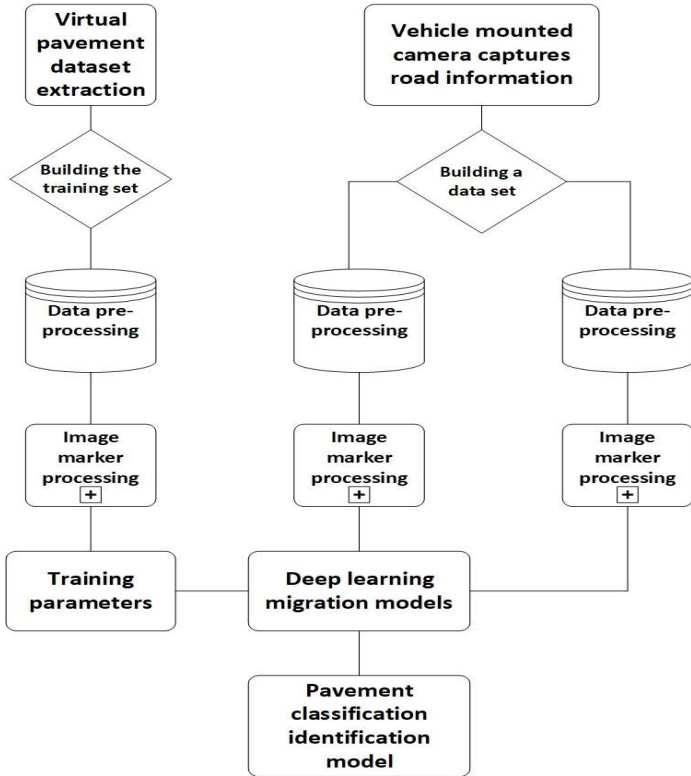

**Figure 6.** The road surface recognition system.

## 3. Results

### 3.1. Data Image Processing

Due to the large size and complex calculation cost of the training dataset required to obtain the vehicle dynamics model, we used a virtual road dataset using high-fidelity data image acquisition. The acquired dataset was enhanced with random inversion, random cropping, random mirroring, and the two-dimensional gamma function proposed in the literature [37], to complete the description of the uniformity calibration process and Gaussian distribution noise [38] on the road dataset. The data preprocessing description and validation set are shown in Table 2. In the image preprocessing stage, the virtual pavement dataset was acquired for random mirroring, cropping, and filling. However, in the dataset, the pavement's state was obscured by some tree guard rails, causing the created dataset to have color shading, which made the different pavement states appear to have similar features and made the convolutional neural network training model much more difficult; therefore, we must perform a calibration process on these datasets. We used the two-dimensional gamma function proposed in the above literature to complete the uniform mapping of the road surface dataset and, finally, used the optimized network structure to classify and output the classes to which the different pavement states belonged. Figure 7 shows the effect of the processed image.

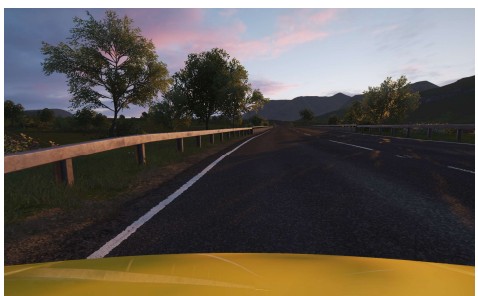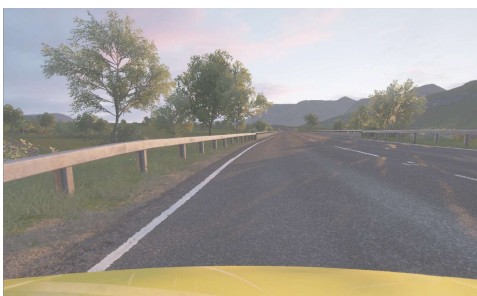

**Figure 7.** Results of shaded light uniformity calibration for the roadway dataset.

**Table 2.** Pavement classification and recognition dataset.

| Type of Pavement | Dataset Display | Number of Images/Times | Number of Training Sets/Times | Number of Validation Sets/Times | Number of Test Sets/Times |
|---|---|---|---|---|---|
| Wet asphalt pavement | | 1200 | 1000 | 500 | 150 |
| Dry asphalt pavement | | 1500 | 800 | 400 | 150 |

**Table 2.** *Cont.*

| Type of Pavement | Dataset Display | Number of Images/Times | Number of Training Sets/Times | Number of Validation Sets/Times | Number of Test Sets/Times |
|---|---|---|---|---|---|
| Snow and ice on roads | | 1700 | 1100 | 600 | 150 |
| Gravel pavement | | 1100 | 900 | 400 | 150 |
| Connecting pavement | | 500 | 200 | 100 | 150 |
| Cement pavement | | 1500 | 1200 | 600 | 150 |

Adding a mean Gaussian distribution noise from 0 to the pixel points on the input images of the dataset is one way to prevent overfitting. Subtle changes to the input images of the training set may degrade the accuracy, so adding noise in the processing of the data images can make the CNN less sensitive to subtle changes in the images and, thus, enhance the generalization ability of the model. We show the results of this treatment in Figure 8.

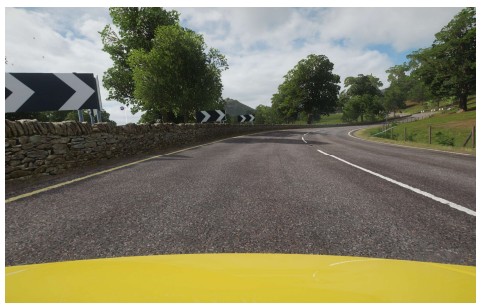
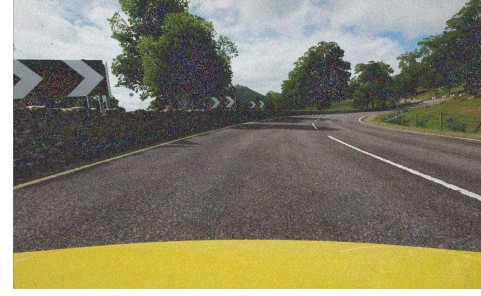

**Figure 8.** Noise-processing results for the road surface dataset.

### 3.2. VGGNet-16 Model Parameter Settings and Training Results

Table 3 shows the parameter settings of the transfer learning model based on VGGNet-16.

**Table 3.** Network model hyperparameter values.

| Parameters | Numerical Values |
|:---:|:---:|
| C | $3 \times 3$ |
| Padding | $2 \times 2$ |
| Stride | 1 |
| Dropout | 0.5 |
| Epoch | 100 |
| Batch size | 8 |
| Adam $(\alpha, \beta_1, \beta_2)$ | $1 \times 10^{-3}, 0.9, 0.9$ |

### 3.3. Pavement Output Results, Based on In-Vehicle Vision Sensors

In this paper, it has been determined that the modified model can be used for pavement classification recognition; therefore, an experimental vehicle was used for pavement acquisition and the final classification was output by the modified VGG-Net-16 transfer learning model. Figure 9 shows the vehicle from which we conducted the pavement experiments, along with the outputs of the pavement classification and CNN.

In summary, the network outputs regarding six kinds of pavement identification and classification are shown, namely, dry asphalt pavement, wet pavement, snowy pavement, concrete pavement, mountain gravel pavement, and buttress pavement. The gravel pavement and buttress pavement are two kinds of complex pavement, and the recognition effect is shown in the above figure. For complex pavement wear conditions, the gravel road surface and connecting pavement images were treated in the pre-processing stage by adding noise due to complex conditions, such as pavement wear and tear, resulting in low recognition accuracy in different weather lighting conditions. The model in the pre-processing training stage acted as the dataset for light-enhancement processing; the first subsection of this chapter has a detailed description of the pre-processing work necessary, while this section shows the output results for the six kinds of pavement. The classification accuracy reached 96.87%. With relatively high accuracy and modest weight space, this model is easier to deploy using domain controllers. This model provides feedforward information about the road surface in front of the vehicle as it travels and can also serve as a reference for additional classification tasks that require real-time feedforward information acquisition.

### 3.4. Accuracy of the Improved Network Model Versus the Original Model

We replaced the original activation function with the Leaky ReLU activation function, residual structure, and dropout layer for the purposes of this paper. A comparison of the accuracies of the two activation functions is shown in Table 4. To verify the high recognition accuracy of the modified VGGNet-16 transfer learning model, different models were tested, and the test results are shown in Table 5. The addition of a residual structure improved the accuracy by 3.43% and 2.79% on the training and test sets, respectively, under the same conditions. The dropout layer that was used improved the accuracy by 3.08% and 3.41%, respectively.

**Table 4.** Comparison of model accuracy by different activation functions.

| Activation Function | Training Set Accuracy | Test Set Accuracy |
|:---:|:---:|:---:|
| Using ReLU activation function | 94.98% | 90.52% |
| Using Leaky ReLU activation function | 98.23% | 94.57% |

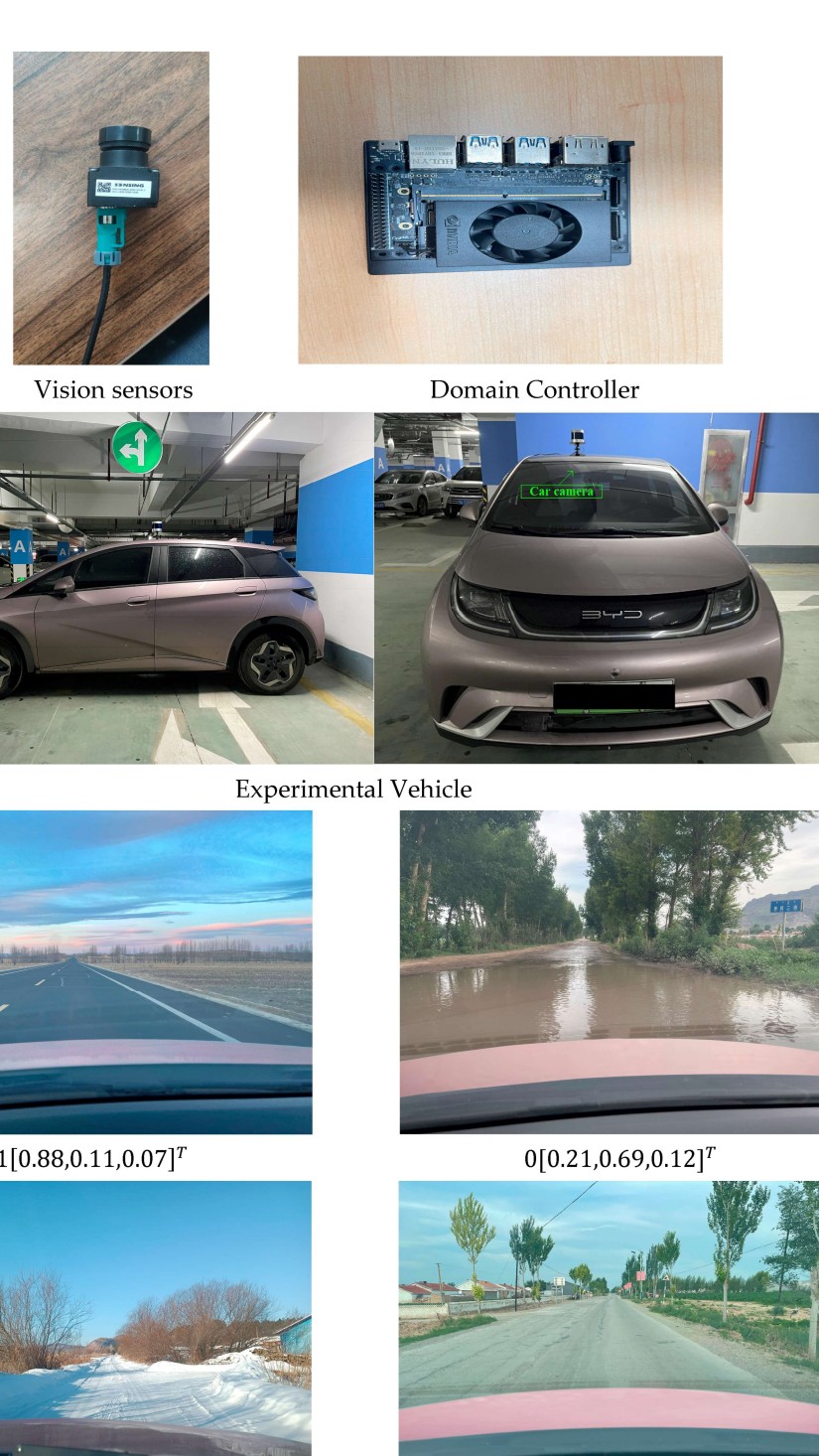

a.           Vision sensors                Domain Controller

b.                          Experimental Vehicle

Output:           $1[0.88,0.11,0.07]^T$                  $0[0.21,0.69,0.12]^T$

$2[0.12,0.12,0.79]^T$                  $3[0.89,0.10,0.27]^T$

**Figure 9.** *Cont.*

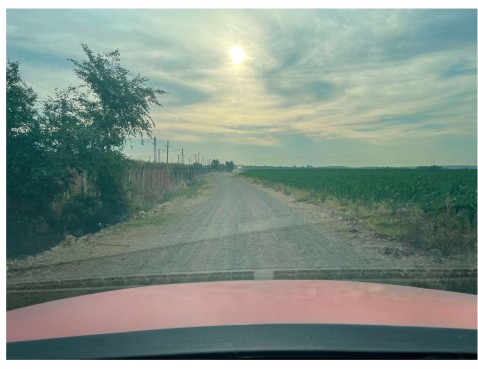

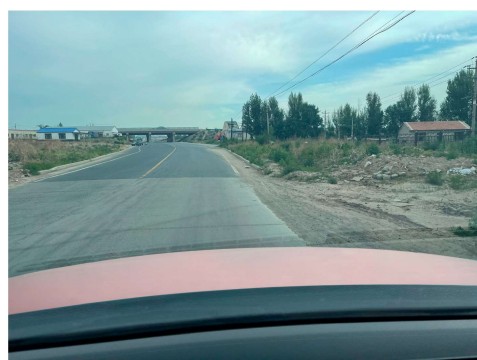

$4[0.94,0.53,0.78]^T$

$5[0.88,0.12,0.19]^T$

**Figure 9.** Output from a pavement classification network based on in-vehicle vision sensors. Instructions: (**a**) Vision sensors and domain controllers. (**b**) Demonstration of installation position of experimental vehicle and vision sensor.

**Table 5.** Performance effects of the original and modified structures.

| Model Structure | Weighted Space | Number of Trainable Parameters | Training Set Accuracy | Test Set Accuracy |
|---|---|---|---|---|
| No dropout layer | 91.3 | 11.6 | 95.17 | 93.46 |
| Residual-free structure | 89.3 | 11.4 | 94.82 | 94.08 |
| Improvements to VGGNet-16 | 91.2 | 11.8 | 98.25 | 96.87 |

*3.5. Model Performance Comparison Test*

In order to further validate the accuracy and recognition of the proposed combined transfer learning and improved VGGNet-16 model for pavement classification presented in this paper, it was compared with four other classical network models under the same test conditions, as well as the training loss curve of the improved VGGNet-16 model. In this paper, as can be seen from Figures 10 and 11, the accuracy curve of the VGGNet-16 transfer learning model (VGGNet-16 transfer) converged after about 45 epochs, con-verging faster among the models compared and allowing less time to train a better model. The loss curve of the model reached 0.4 after 100 epochs and the LOSS curve steadily decreased as the number of training batches increased. In this paper, the dropout layer was added to the model improvement, and the curves show that the improved VGGNet-16 model curve was relatively stable.

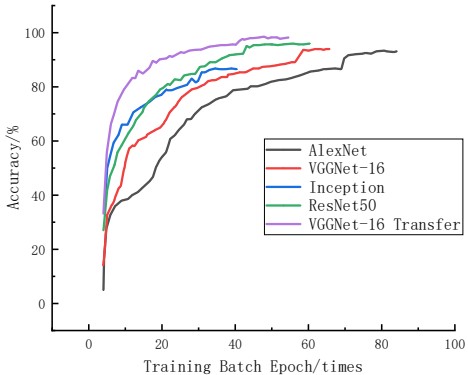

**Figure 10.** Graph showing 5 network recognition accuracy curves.

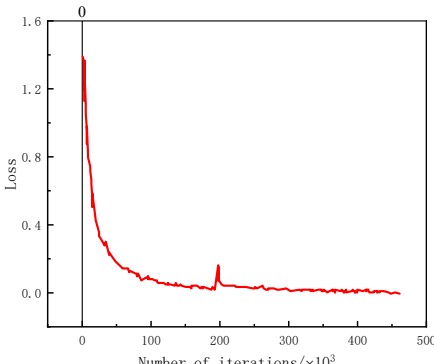

**Figure 11.** VGGNet-16 transfer training LOSS curve.

*3.6. Model Performance Comparison Test*

To further illustrate the effectiveness of the improved model based on VGGNet-16 combined with transfer learning for classifying pavement discrimination built in this paper, it was compared with four different classical convolutional neural networks under the same dataset and hardware configuration, and the accuracy is shown in Figure 9 above. In the model validation accuracy curve, the improved transfer learning model VGGNet-16 in this paper has a gradual smoothing trend after 45 epochs, and its convergence speed was faster compared with the other four models. It is also possible to train a more efficient model in a relatively short period of time.

As shown in Table 6, the VGGNet-16 transfer incorporates the residual structure and Leaky ReLU activation function with better performance, thus achieving 98.25% and 96.87% accuracy in the training and validation sets, respectively, compared to AlexNet, ResNet50, InceptionV3, and VGGNet-16. The accuracies of 98.25% and 96.87% on the training and validation sets, respectively, are 1.5, 0.4, 2.08, and 0.68 higher than those of AlexNet, ResNet50, InceptionV3, and VGGNet-16, respectively, indicating that the pavement classification is better than that of the other models. In terms of the weight space and the number of training parameters, the VGGNet-16 transfer model does not stack too many layers to improve the accuracy, but freezes the weights, performs transfer learning, and changes the parameters of the fully connected layer neurons for training; therefore, the weight space of VGGNet-16 transfer is 91.2 MB.The average training time for the validation set of the VGGNet-16 transfer model is 51.86 ms. The Adam optimizer chosen for this optimization is a gradient-descent algorithm-based optimizer that combines the ideas of momentum method and adaptive learning rate and can converge more rapidly onto the optimal solution. The core idea of the Adam optimizer is to optimize the model more efficiently during training by updating the parameters, based on the first-order moment estimates and second-order moment estimates of the gradients, while adaptively adjusting the learning rate. Compared to the original VGGNet-16 model, its accuracy, weight space, and average time are all improved.

**Table 6.** Comparison table of model parameters.

| Five Network Models | Weighted Space | Number of Training Parameters | Training Set Accuracy/% | Test Set Accuracy/% | Time Taken for Test/sec |
|---|---|---|---|---|---|
| AlexNet | 164 | 21.6 | 95.28 | 95.37 | 63.17 |
| ResNet50 | 180 | 23.5 | 97.46 | 96.47 | 58.15 |
| InceptionV3 | 167 | 21.8 | 91.19 | 94.79 | 52.84 |
| VGGNet-16 | 256 | 33.6 | 95.83 | 96.19 | 56.74 |
| VGGNet-16 transfer | 91.2 | 11.8 | 98.25 | 96.87 | 51.86 |

In summary, the proposed VGGNet-16 model combined with transfer learning, i.e., VGGNet-16 transfer, exhibited the best comprehensive performance for pavement classification and recognition, and is, thus, more suitable for deployment in domain controllers or

other devices than other classical networks. The speed can meet the real-time requirements for pavement recognition and the reliability and adaptability are also better.

## 4. Conclusions

(1) In this paper, by introducing the Leaky ReLU activation function to replace the activation function in the original model, adding three residual structures and two dropout inactive layers to improve the VGGNet-16 model with parameter adjustment, we propose a model combining transfer learning and the improved VGGNet-16 model for pavement classification recognition. After extensive training, the accuracy of recognition can reach 96.87%, which effectively solves the problem of low efficiency and accuracy of traditional machine learning for pavement recognition; it senses the type of pavement in advance and plays an important early warning role for safety.

(2) In this paper, five models are compared; the improved VGGNet-16 model shows a larger improvement in accuracy compared to the remaining four models, but there is still some room for improvement in terms of training time. At the same time, this paper provides poorer results on the training set for the recognition of muddy roads or unpaved roads, which is a direction to explore to achieve continued optimization of the network.

To add to the second point above, in this paper, 6 kinds of road surface recognition are employed in the dataset training. However, non-paved muddy road recognition is not very effective at the data processing level, due to inconsistent light exposure conditions. With the use of data light enhancement, we know that the vehicle vision sensor used to collect information will have the biggest impact. The first issue is that the light is uniformly different, but in the muddy road data by light enhancement, the image is still unclear. The light distribution is not uniform; if the same comparison of the image in the dark were made instead, the recognition would be very good. We also utilized different sections of muddy roads for surface recognition. However, in the collected image data, the color difference between the muddy road and the outside world is not obvious and there are few rutted tracks. The recognition effect would be better when the muddy road contains many rutted tracks and the road color difference from the surroundings is more obvious. In terms of the recognition of road surface, there are two typical errors regarding these features: the first is that the difference between the recognized road surface color and the surrounding color is great, even after pre-processing, but there are still deviations that will lead to recognition error. The second is that the identification of individual unpaved surfaces will be disrupted by numerous elements, including ruts, pavement integrity issues, and additional damage to the original fabric of the pavement, which will lead to identification errors. By increasing the number of layers of the network, it should be possible to achieve improved recognition accuracy; however, increasing the number of layers will increase the weight of the network and the average training time, which is not in line with the characteristics of real-time recognition, accuracy, and efficiency. This will be addressed in follow-up studies. Other researchers may wish to address the problem of finding methods to perform accurate recognition for unpaved complex pavements without changing the model.

(3) Future work will combine the method proposed in this paper with the vehicle-based three-degree-of-freedom dynamics simulation to verify the adhesion coefficient of the driving road surface, propose a fusion strategy based on dynamics information and visual information, and design a study on the recognition of early warning cues and the recognition of road surface types and adhesion coefficients.

To add to the third point above, the main aim of this paper was to use the vehicle vision sensor to pre-empt perception of the vehicle driving road surface, but the accuracy of this perception only reaches 96.87%, and cannot reach 100% accuracy regarding recognition of the road surface. However, combined with the three-degree-of-freedom dynamic vehicle model, the system can use the real-time data of the wheel speed sensor through the Carsim

program to verify the current road surface adhesion information visually. The fusion strategy of visual and kinetic information can verify the adhesion coefficient of the road surface more accurately, which is estimated using a single method in the previous literature; thus, there is a relatively large error and the estimation does not happen in real time. The visual information has the advantage of being taken in real time and can be used as feed-forward information in the fusion strategy, while the kinetic information can be used as back-end validation information. In future work, this can be used to estimate the slip rate and adhesion coefficient of the feed-forward information and can also correct the feed-forward information.

**Author Contributions:** J.Z., methodology, writing—original preparation, W.G., methodology, visualization, resources, investigation, software, F.W., data curation, writing—review and editing. All authors have read and agreed to the published version of the manuscript.

**Funding:** This research was funded by the Key Research and Development Project of Hubei Province (2021BAA180), the Project of National Natural Science Foundation of China (52202480), the Project of Natural Science Foundation of Hubei Province (2022CFB732), the Project of Natural Science Foundation of Hubei Province (2020CFB119), the Scientific Research Program Guidance Project of Hubei Provincial Education Department (B2021008).

**Data Availability Statement:** Not applicable.

**Conflicts of Interest:** The authors declare no conflict of interest.

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
