# Peer review of "A Study on Pavement Classification and Recognition Based on VGGNet-16 Transfer Learning"

_electronics, doi:10.3390/electronics12153370_

Round 1
Reviewer 1 Report
This paper, electronics-2501593, presents an in-depth study on the implementation and improvement of a VGGNet-16 based deep learning model for pavement classification and recognition. The objective is to enhance the capabilities of intelligent networked vehicles by enabling real-time and accurate identification of road surfaces. The authors have done an admirable job of explaining the need for such a system in the context of rising automobile traffic and associated safety concerns. They have also highlighted the limitations of traditional image recognition and classification methods, which suffer from low accuracy and adaptability. The methods section is comprehensive and well-structured, detailing the improvements made to the original VGGNet-16 model. The inclusion of the Leaky ReLU activation function, three residual structures, and two Dropout layers is a noteworthy modification. The authors demonstrate a clear understanding of the model, evident from their detailed and lucid explanations. The paper is particularly strong in its experimental results section, providing clear comparisons between the improved VGGNet-16 model and competing models like AlexNet, InceptionV3, and ResNet50. The reported 96.87% classification accuracy of the proposed model is impressive, suggesting its potential utility in real-world applications. One area where the paper could be improved is in its discussion of the limitations and potential drawbacks of their model. While the model clearly outperforms others, it would have been useful to see a more thorough discussion of its limitations, possible shortcomings, and potential mitigation strategies. Further, while the study recognizes the need for large datasets for model training, it doesn’t provide enough details about the scale and diversity of the dataset used for their model. The proposed future work, including integrating the developed model with vehicle-based three-degree-of-freedom dynamics simulation, is intriguing and shows a promising direction. Overall, this paper is a significant contribution to the field of intelligent networked vehicles. It provides a robust solution to a pressing problem and presents an effective approach that leverages deep learning and transfer learning. The research has a high potential for practical applications and can significantly influence future work in this domain. However several things need to be clarified, I would recommend a major revision and I will ask the authors to answer the questions below.
General comments:
- The paper successfully addresses a critical challenge in the field of intelligent networked vehicles - the real-time and accurate classification and recognition of pavements.
- The authors effectively improved the VGGNet-16 model by introducing the Leaky ReLU activation function, adding three residual structures, and incorporating two Dropout layers.
- The experimental section is particularly strong, featuring comprehensive comparisons with competing models, yielding impressive results with a reported classification accuracy of 96.87%.
- The paper could benefit from providing more information about the dataset used, including collection and pre-processing methods. This addition would enhance the transparency and reproducibility of the study.
- The proposed future work on integrating the developed model with a vehicle-based three-degree-of-freedom dynamics simulation is an interesting direction that could further enhance the system's utility.
- Overall, the paper is a significant contribution to its field, offering a robust solution leveraging deep learning and transfer learning for a critical issue in intelligent networked vehicles.
Questions for authors:
1. Could you provide more information about the dataset used in your study, including the method of collection and the pre-processing techniques utilized? This would be helpful for the reproducibility of your research.
2. Have you considered testing your model in more diverse conditions, such as different lighting, weather, or geographic variations?
3. You mentioned replacing the ReLU activation function with the Leaky ReLU activation function. Could you discuss more about the rationale and benefits of this modification?
4. What are the specific improvements that the three residual structures and two Dropout layers bring to the model's performance? Could you elaborate on this?
5. Your future work involves integrating the model with a vehicle-based three-degree-of-freedom dynamics simulation. Could you share more about this planned integration and its potential benefits?
6. In your comparison of the VGGNet-16 model with other models (AlexNet, InceptionV3, and ResNet50), how did the models perform in terms of computational resources and processing time?
7. Could you explain how your model can handle the pavement changes that happen over time due to wear and tear, or seasonal changes (like snow or leaves on the road)?
8. Are there any plans to make your model adaptable or scalable to other applications beyond pavement recognition in intelligent networked vehicles?
9. The 96.87% accuracy achieved is indeed impressive. However, could you discuss more about the instances where the model failed to correctly classify the pavement? What were the typical characteristics of these misclassified examples?
The English language quality of the paper is generally good, but several areas could be improved. The paper has an overuse of the passive voice, which can make the text harder to read and understand. Using more active voice could improve clarity. Some sentences are excessively long and contain multiple ideas. Breaking these down into shorter, more focused sentences would enhance readability. The overuse of conjunctions, like 'and' and 'while,' leads to run-on sentences. Simplifying these sentences could improve readability. The paper also has some inconsistency in the use of terminology, such as the different ways the VGGNet-16 model is referred to. Consistent use of terms would improve clarity and professionalism. Finally, the use of the term "migration learning" is less common than "transfer learning" in machine learning literature. Adhering to common terminology can enhance the clarity and accessibility of the paper. While the overall quality of English appears good, minor revisions could improve the readability and accessibility of the paper.
Reviewer 2 Report
This manuscript simply applied the deep learning classification approach for pavement classification. While the engineering practice purpose is not clear? Will a GPS & GIS system help you map the pavement issues? The authors should provide a completed workflow to use the proposed method, not just apply the classification.
In addition, please improve the literature review. There are some ASCE publications as references. Many deep learning papers have been published in the Journal of Computing in civil engineering, the Journal of construction engineering and Management, the Journal of Performance of constructed facilities, and the Journal of transportation engineering Part B Pavements.
Reviewer 3 Report
The method proposed by the authors is interesting and is of interest to this field.
The elements presented by the authors are well founded, and the obtained results confirm to a good extent that the proposed method represents an improvement.
In the conclusions chapter, in paragraph 2, the authors add some limitations of the proposed method. In relation to this aspect, I propose the authors to make an addition regarding the reasons why the method obtains poorer results for unpaved roads.
Figure 2 has a typo.
In the reference list, the authors forgot to specify the Day, Month and Year for some references.
A minor observation is related to chapter 5. I believe that the authors should not consider this information as a chapter. The paper template of MDPI's Electronics magazine allows such information to be provided in another way.
Reviewer 4 Report
The work presents an interesting deep learning-based method for classifying and identifying road surfaces, which uses an improved (VGGNet-16) model combined with a migration learning approach to acquire the road surface in front of the vehicle through an onboard camera.
In general, the work presents adequate mathematical support and correct experimental development. However, the only suggestion goes in terms of the conclusions. These must be structured without numerals and appropriately, in addition to including hard and relevant data results.
Check some small typos and grammatical errors
Round 2
Reviewer 1 Report
Upon careful examination, I can confirm that the necessary revisions have been duly addressed and made by the authors in response to the comments and suggestions given in the previous review. The manuscript now meets the standards required for publication.
I can confirm an improvement in the standard of English.
Reviewer 2 Report
The authors did not answer the questions raised by the reviewer.
Round 3
Reviewer 2 Report
Still recommend the authors to address the engineering practice.
